# Stem Anatomy Confirms *Tingia unita* Is a Progymnosperm

**DOI:** 10.3390/biology12040494

**Published:** 2023-03-24

**Authors:** Yang Yang, Shi-Jun Wang, Jun Wang

**Affiliations:** 1State Key Laboratory of Palaeobiology and Stratigraphy, Nanjing Institute of Geology and Palaeontology and Center for Excellence in Life and Paleoenvironment, Chinese Academy of Sciences, No. 39 East Beijing Road, Nanjing 210008, China; 2University of Chinese Academy of Sciences, No. 19 (A) Yuquan Road, Shijingshan District, Beijing 100049, China; 3State Key Laboratory of Systematic and Evolutionary Botany, Institute of Botany, Chinese Academy of Sciences, Beijing 100093, China

**Keywords:** stem, anatomy, scalariform bordered pits, Noeggerathiales, early permian

## Abstract

**Simple Summary:**

The systematic position of Noeggerathiales was long uncertain until the whole plant species *Paratingia wuhaia* was restored and proved to belong to progymnosperms due to its spore-producing fertile organ and secondary wood producing a large stem. However, whether *Tingia* Halle as the most diversified genus in Noeggerathiales belongs to progymnosperms has yet remained uncertain as the anatomy of the main stem of this plant is unknown, that is, whether the anatomy of the main stem with the anatomical characteristics of gymnosperms woods remains uncertain. Here, the stem anatomy of *Tingia unita* is presented based on fossil materials from the early Permian Wuda Tuff Flora in Wuda Coalfield, Wuhai City, Inner Mongolia, China. The well-developed secondary wood, plus previously accumulated evidence of the spore-bearing nature of this plant, confirms that this genus belongs to progymnosperms. As such, the genera *Tingia* and *Paratingia* are all certainly progymnosperms in affinity.

**Abstract:**

*Tingia* Halle, a representative genus of the Cathaysia Flora, has been studied for nearly 100 years, being a small heterosporous tree based on the gross morphology of *Tingia unita*. However, the systematic affinity of *Tingia* is uncertain. Now, a number of well-preserved fossils of *T. unita* from the Taiyuan Formation of Lower Permian in Wuda Coalfield, Wuhai City, Inner Mongolia facilitates an examination of wood anatomy. The stem anatomy of *T. unita* shows parenchymatous pith, endarch primary xylem, pycnoxylic secondary xylem, and cortex, typically a type of gymnosperm wood, which taken together with pteridophytic reproduction, certainly evidences that *Tingia* Halle is a progymnosperm. In addition, *Tingia* together with *Paratingia* provide strong evidence to link the Noeggerathiales with progymnosperms.

## 1. Introduction

Well-preserved fossil woods can provide valuable anatomical information not only on the wood structure, habit, but more significantly on the botanical affinity, in addition to their palaeoecological, palaeoenvironmental, and palaeoclimatological implications where they grew [1,2,3,4].

The order Noeggerathiales is an enigmatic and extinct plant group in the Late Palaeozoic on Cathaysia and Euramerica continents. Since it was first proposed in 1931, around 50 species of 20 genera have been enrolled [5,6,7]. However, the systematic affinity of this group was poorly understood until the stem anatomy of *Paratingia* Zhang was studied [7]. Up to now, growth habit has been reported for only four species, namely *Noeggerathia foliosa* [8], *Paratingia wudensis* [9], *Paratingia wuhaia* [7], and *Tingia unita* [10,11], of which only *Paratingia wuhaia* is known for stem anatomy [7], and accordingly it has been concluded that Noeggerathiales belongs to progymnosperms. However, there is no information from the stem anatomy of other taxa in Noeggerathiales, to verify the progymnosperms affinity of this group.

Here, three partially permineralized specimens were studied to reveal the anatomy of the stems of *Tingia unita* Wang, which consists of pith, primary xylem, pycnoxylic secondary xylem, and cortex. These new features confirm that *T. unita* is also a progymnosperm combined with previously known characteristics [10,11]. This further strengthens the systematic relationship between Noeggerathiales and progymnosperms.

## 2. Materials and Methods

Five specimens here were collected from a volcanic tuff bed in the uppermost Taiyuan Formation of early Permian in Wuda Coalfield, Wuhai City, Inner Mongolia. The peat-forming vegetation preserved in the volcanic tuff between coal seams No.6 and No.7 is also known as “vegetational Pompeii” [12,13,14,15] (Figure 1). High-precision U–Pb dating of zircons established the age of the tuff as 298.34 ± 0.09 Ma in the earliest Permian [16]. Through the years, fossil specimens of *Tingia unita* have been not only found over the outcrops here and there randomly in the coalfield, but also more systematically collected from those sites (Figure 1C) where quantitative documentation has been conducted. In particular, the material with the preservation of stem anatomy was collected during a quantitative documentation of the peat-forming forest by quadrat method (1 m × 1 m) (Figure 1E) in 2021.

The specimens were first photographed with a Canon Power Shot G7 X Mark III in the field. Subsequently, macroscopical images were taken indoors using a Nikon D800 camera (Figure 2 and Figure 3). Materials with anatomy were studied by thin sections using the technique reported by Hass and Rowe [17]. Thin sections were photographed using stereoscope Zeiss Axio Zoom V16 and microscope Zeiss Axio Imager Z2. All measurement data were measured with Image J. Specimens (PB201150–PB201152, PB201694–PB201695) and all slides (PB201150-1 to PB201150-25, PB201151-1 to PB201151-5, PB201152-1 to PB201152-22) are stored in the palaeobotanical collection of Nanjing Institute of Geology and Palaeontology, Chinese Academy of Sciences.

**Figure 1 biology-12-00494-f001:**
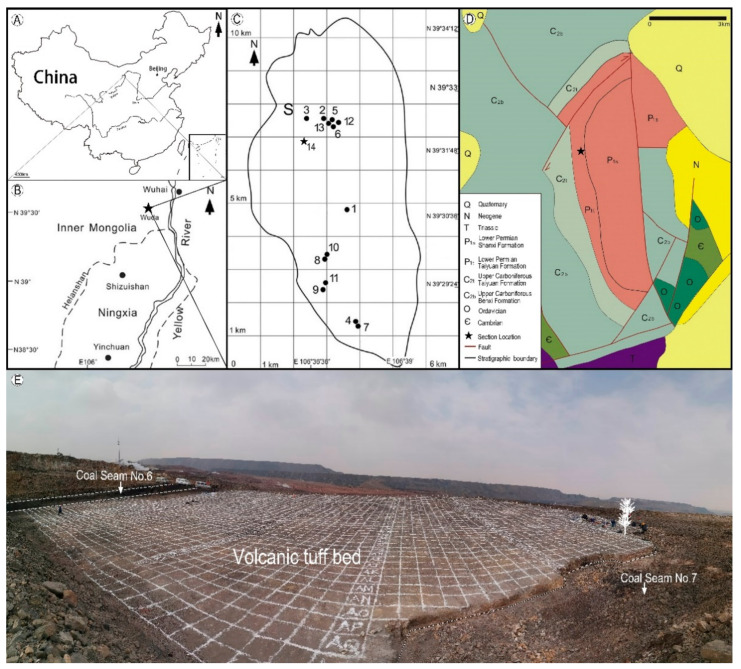
The location and stratigraphic horizon of study area. (**A**–**C**) The locality of the fossil materials, Wuda Coalfield, Inner Mongolia, China (No. 1–14 indicate locations where quantitative investigation on this peat-forming flora were conducted, of which No. 14 black star indicates the present fossil collection site) (modified from Wang et al. [14]). (**D**) Geological map of the Wuda Coalfield (black star indicates collection site of the present materials) (modified from He et al. [15]). (**E**) Bedding plane of the tuff bed between coal seams No. 6 and No. 7, on which quadrat 1 m × 1 m has been outlined, with two tree crowns indicating where the specimens were collected.

## 3. Results

### 3.1. Systematics

Order: Noeggerathiales Němejc, 1931 emend. Wang et al., 2021

Family: Tingiostachyaceae Gao et Thomas, 1987 emend. Wang et al., 2021

Genus: *Tingia* Halle, 1925.

Species: *Tingia unita* Wang, 2006 emend. Yang et al.

Emended diagnosis: The once-pinnate compound leaves and strobili directly arising from the monopodial stem. Pinnules of the compound leaf are plagiotropically attached on the rachis and arranged in four rows. Among them, two rows of large pinnules are on the abaxial side of the rachis and they are broadly oblong to linear. The apex of large pinnules is wedge-shaped and dentate, the lateral margin is entire. The veins of large pinnules are dichotomize at the base part, then run parallel to each other and extending to the apical lobes. Two rows of small pinnules are on the adaxial side of the rachis and they are linear. Rachis of the compound leaves with a C-shaped or U-shaped vascular bundle in cross section. Strobili are cylindrical in shape, heterosporous, and bisporangiate. The stem is eustelic and consists of pith, endarch primary xylem, pycnoxylic secondary xylem, and cortex. Tracheids of secondary xylem have uniseriate scalariform bordered pits and few uniseriate oval bordered pits on the radial walls. Each cross-field has several scalariform, sub-circular or oval bordered pits. Rays are normally uniseriate and low. Cortex consists of parenchyma cells. Leaf traces are fan-shaped or tangentially elongated.

Remarks: The species *Tingia unita* was established for a small tree with the compound leaves and strobili directly attached on the top part of same monocaulus stems. Here, the main work of this manuscript is to report the anatomical information of the stem. In order to avoid the introduction of superfluous names, the specific diagnosis of *T. unita* is emended to include the stems, the vegetative and the fertile organs of the same plant.

Materials: fossil wood (PB201150 to PB201152), Fossil leaf and strobili (PB201694 to PB201695) and slides (PB201150-1 to PB201150-25, PB201151-1 to PB201151-5, PB201152-1 to PB201152-22).

Repository: The palaeobotanical collection of Nanjing Institute of Geology and Palaeontology, Chinese Academy of Sciences.

Type locality: Wuda Coalfield, Wuhai City, Inner Mongolia, China.

Stratigraphic horizon: The upper part of the Taiyuan Formation

Age: The earliest Asselian, Permian

### 3.2. Description

The stem is monopodial and unbranched with a crown composed of once-pinnate compound leaves and strobili, both directly arising from the stem (Figure 2A). The detailed description of the compound leaves and strobili in Wang [10] and Wang et al. [11] is still effective. These stems were severely flattened and only partly with preservation of anatomy. The major diagnostic features are yielded from the fossil stems PB201150 and PB201151 (Figure 2D and Figure 3C), which bear directly attached compound leaf and strobilus, and the fossil stem PB201152 (Figure 3E) adds better illustration of the pith and secondary xylem. The stem is oblate oval with a maximum diameter of ca. 14 mm × 64 mm in the cross section, eustelic, consisting of pith, endarch primary xylem, pycnoxylic secondary xylem, and cortex (Figure 4A).

#### 3.2.1. The Compound Leaf and Strobilus

The leaf is once pinnate compound leaf which is attached directly to the stem (Figure 2A). Pinnules of the compound leaf are plagiotropically arising from rachis (Figure 2B,C and Figure 3B). Among them, the large pinnules are linear, with entire lateral margins. The apex of the large pinnules is wedge-shaped and finely dentate. The veins of large pinnules are dichotomized at the base part, then running parallel to each other and extending to the apical lobes. The small pinnules are linear (Figure 2C). The rachis is oblate in cross section, with a C-shaped vascular bundle (Figure 2E). Strobili are cylindrical in shape and they are directly attached to the stem by a long stalk (Figure 2D and Figure 3B–D). Strobilus are heterosporous and bisporangiate (Figure 2F,G). The shape of sporangia is an ellipsoid. Microspores are circular to triangular, with trilete mark (Figure 2G).

#### 3.2.2. Stem Anatomy

##### Pith

In cross section, the pith is elliptical in shape and its maximum size is about 10 mm × 56 mm, and it consists of parenchyma cells and secretory canals (Figure 4). The parenchyma cells are sub-circular, elliptical, oval to irregular in shape (Figure 4B,D). The radial diameter of these parenchyma cells varies from ca. 22 μm to 303 μm and their tangential diameter varies from 17 μm to 220 μm. There is a narrow band consisting of irregular-shaped parenchyma cells, which divides the pith into two parts (Figure 4B,E). Secretory canals are randomly distributed at the outside of the pith, and they are elliptic in shape with the size of ca. 97 μm × 149 μm to 140 μm × 340 μm in cross section (Figure 4B,C), and the length can be up to 3748 μm in radial section (Figure 4D).

##### Primary Xylem

In cross section, the primary xylem is indistinct due to poor preservation. According to the specimens, the primary xylem could be endarch (Figure 5A,B). However, it is hard to determine protoxylem, metaxylem, and their size due to the limitations of preservation. The primary xylem is triangular (Figure 5A) or C-shaped (Figure 5B). In radial section, the tracheids of primary xylem have helical thickenings on the walls (Figure 5C,D).

##### Secondary Xylem

The pycnoxylic secondary xylem consists of tracheids and rays, with a maximum width of ca. 6 mm in the cross section (Figure 4A and Figure 5E–G). Tracheids are rectangular and polygonal in shape, with a radial diameter varying from 9 μm to 55 μm (*n* = 899, mean = 28 μm). Tracheid diameter is basically less than 50 μm (>98%). Rays are parenchymatous, and the tangential width of ray cells is usually narrower than the tangential width of tracheids in cross section. Occasionally, the tangential width of some ray cells is wider than the tangential width of adjacent tracheids (Figure 5E,G). In addition, the tangential diameter of tracheids of partly secondary xylem is longer than the radial diameter of tracheids after compressing along the radial direction (Figure 5G). Moreover, this squeezing shortens the radial diameter of the ray cells, which causes the thick-walled appearance of ray cell ways (Figure 5G).

In radial section, the radial tracheid walls basically possess uniseriate scalariform bordered pits and few uniseriate oval bordered pits (Figure 6A–E). In cross-field, there are normally 3–8 scalariform pits arranged in one-column, few mixed type with scalariform, sub-circular and oval pits, or biseriate alternately elliptical pits (Figure 6F–H). However, it is hard to determine whether the cross-field pits are bordered or simple due to the limitations of preservation. Ray cells are brick-shaped with straight horizontal walls and vertical or oblique tangential walls (Figure 6I), and sized ca. 26 μm to 141 μm in length and ca. 20 μm to 57 μm in height.

In tangential section, the end walls of tracheids are gradually tapered (Figure 7B). The shape of the ray cells varies from rectangular, triangular to polygonal. Rays are usually fusiform. Sometimes, the ray cells in the middle part of rays are smaller than the ray cells on both sides of the same rays (Figure 7C). Rays are usually fusiform, few partly biseriate, varying from 1 to 26 (typically 1–8) cells high (*n* = 790, mean = 4) (Figure 7A–E and Figure 8). The height of the ray cells varies from 12 μm to 76 μm (*n* = 208, mean = 38 μm), and the tangential width of the ray cells varies from 11 μm to 44 μm (*n* = 208, mean = 27 μm). Pitting on tangential tracheid walls are absent by the critical examination of all the slides.

##### Cortex

The cortex has a maximum width of ca. 12 mm and consists of parenchyma cells (Figure 4A and Figure 9A–E). In cross section, the shape of parenchyma cells varies from circular, oval, triangular, hexagonal to polygonal. Their radial diameter varies from 38 μm to 300 μm. Usually, the parenchyma cells in the inner region of cortex are more fragmented, while they are more intact in the outside of the cortex (Figure 9A). In radial section, the shape of parenchyma cells varies from oval, polygonal to irregular-shaped, and they are usually longitudinally elongated (Figure 9C,D).

##### Leaf Trace

Leaf traces are fan-shaped or band-like (tangentially elongated) when they occur in the cortex (Figure 9E–I). Secondary xylem is observed at the abaxial side of the leaf traces (Figure 9G).

## 4. Comparison and Discussion

### 4.1. Comparison with Tingia unita and the Relationship among These Stems

It is worth mentioning that specimen PB201150 and specimen PB201151 are directly attached with the pinnules and strobilus. Among them, the critical features of our specimens are both anisophyllous and dorsiventral, and pinnules possessing parallel venation. Moreover, the large pinnules are linear with a wedge-shaped apex and entire lateral margins, and the small pinnules are also linear but narrower than the large pinnules. All of these confirm the identical characteristics of *Tingia unita* [10,11]. So, the specimens PB201150 and PB201151 belong to *T. unita*.

The wood structure of stem can be divided into pycnoxylic wood and manoxylic wood, while it is difficult to define them satisfactorily through visual observation solely [18]. Here, a relatively reliable combination of quantitative features of pycnoxylic and manoxylic secondary xylem has been accepted by many authors [19,20,21,22]. Namely, tracheids of pycnozylic secondary xylem are smaller than 50 μm in diameter, and its rays are both uniseriate and short, that is well recognizable in the wood of conifers. The manoxylic secondary xylem has a large diameter of tracheids (up to 150 μm or more), and rays are broad (more than 5 rows) and tall (up to 200 cells), as the wood of cycads [19,20,21,22]. Considering that both pycnoxylic and manoxylic wood are found in different species of the same genus, e.g., *Pitus*, pycnoxylic and manoxylic wood have little taxonomic significance due to their difference [18].

In our specimens, both of the diameter of secondary xylem tracheids (9 μm to 55 μm, 98% < 50 μm) and rays (uniseriate and short) are well accordant with the definition of pycnoxylic wood, therefore, the stem of *Tingia unita* has pycnoxylic wood anatomy. Although, the specimen PB201152 has no organic attachment of leaf or strobilus of *T. unita*, it also can be assigned to *T. unita* due to its close similarity to the other two specimens (PB201150 and PB201151). They are all the same in having pycnoxylic wood; uniseriate scalariform bordered pits in radial tracheid walls of secondary xylem; scalariform, oval, and mixed type pits in cross-field; uniserial and low rays. Although there are some slight differences between these three stems. For example, specimen PB201152 has secretory canals in pith, whereas specimen PB201150 has only several cavities with secretory substance, and the uniseriate oval bordered pits of specimen PB201150 is slightly more obvious than specimen PB201152. However, the pith and secondary xylem of specimen PB201152 are more completely preserved than those of specimen PB201150 and PB201151, and the uniseriate oval bordered pits only appear on the narrower tracheid walls. So, all of these fossil stems can be assigned to *T. unita*.

### 4.2. Comparison with Species from the Same Collection Area

None of the associated taxa found in this area, including *Sigillaria*, *Asterophyllites*, *Sphenophyllum*, *Pecopteris*, *Sphenopteris*, *Alethopteris*, *Taeniopteris,* and *Pterophyllum*, representing lycopsids, sphenopsids, marattialean ferns, seed fern, cycads, or bennettitaleans, possess anatomical similarity with present specimens. In particular, differing from the present stem, *Sigillaria* possesses a smaller vascular cylinder relative to the diameter of the whole stem [21,23,24]; *Asterophyllites* has pith surrounded by a number of collateral vascular bundle and each with a conspicuous protoxylem canal (carinal canal) [21]; *Sphenophyllum* has a protostelic vascular cylinder with triangular to subtriangular primary xylem in outline; Marattialean fern with pecopterid pinnules is arborescent and has a stem composed of a series of concentric amphicrybral cauline bundles separated by leaf gaps, and enclosed by numerous adventitious roots (root mantle) [21,25]. *Sphenopteris* and *Alethopteris* are respectively a herbaceous and semi-self-supporting plant [12,26] and do not have a thick major stem like the specimens in this study. *Taeniopteris* and *Pterophyllum* are cycadean or bennettitaleans in affinity, with their major stem producing manoxylic wood based on our preliminary examination.

### 4.3. Comparison and Discussion with Other Species

The most remarkable feature of *Tingia unita* is that the radial tracheid walls of the secondary xylem has basically uniseriate scalariform bordered pits. In the Paleozoic, the scalariform bordered pits on radial walls of the pycnoxylic secondary xylem were reported from: two putative progymnosperm *Paratingia wuhaia* [7] and *Protopitys buchiana* [27,28,29,30], a pteridophyte *Johnhallia lacunosa* [31], two coniferophyte *Xuanweioxylon scalariforme* [32] and *Xuanweioxylon damogouense* [33], a putative Cordaitalean *Cordaites missouriense* [32,33,34] and a gymnosperm *Yangquanoxylon miscellum* [35]. Although there are some other species which also possessing scalariform bordered pits, they are only found in the transition zone between metaxylem and secondary xylem formed at the earliest stage, such as *Protopitys scotica* [30,36,37] and *Guizhouoxylon dahebianense* [38].

*Paratingia wuhaia* is similar to *Tingia unita* in possessing uniseriate scalariform bordered pits on the radial tracheid walls of secondary xylem and uniseriate rays. However, the rays (1–26 cells high) of *T. unita* are taller than those of *P. wuhaia* which are only 1–7 (typically 1–4) cells high. In addition, there are a few uniseriate oval bordered pits on the radial tracheid walls of secondary xylem of *T. unita*, which has not been reported from *P. wuhaia*. Finally, data on pith and cross-field pits of *P. wuhaia* are unknown which limits further comparison with *T. unita*.

*Protopitys buchiana* possessesendarch primary xylem, pycnoxylic secondary xylem, uniseriate scalariform bordered pits on the radial tracheid walls and uniseriate rays, which are similar to *Tingia unita*. However, the primary xylem of *Protopitys* is distributed at both ends of the pith and forms two opposite zones that distinguishes it from *T. unita*. Moreover, the rays of *P. buchiana* are generally less than ten cells in height, clearly shorter than *T. unita*. In addition, the cross-field pits of *P. buchiana* are also different from *T. unita*, the former possesses 8 to 20 simple circular to elliptical pits, while the latter possesses mainly scalariform pits.

Differing from *Tingia unita*, *Johnhallia lacunosa* has scalariform and multiseriate oval to circular bordered pits on the radial tracheid walls of secondary xylem, and tangential tracheid pits on secondary xylem, as well as five independent protoxylem poles.

*Xuanweioxylon scalariforme* and *X. damogouense* are similar to *Tingia unita* in possessing uniseriate scalariform bordered pits on the radial tracheid walls of secondary xylem and uniseriate rays. However, the cross-field pitting of *Xuanweioxylon* is different from *T. unita*, namely, *X. scalariforme* possesses 1–2 large, oval, or nearly round simple pits, or several small oculipores; *X. damogouense* has 3–19 circular, oval, polygonal, and scalariform bordered pits. Moreover, there are uniseriate to multiseriate oval to circular bordered pits on the radial tracheid walls of *Xuanweioxylon*, also distinguishing it from *T. unita*.

*Cardaites missouriense* is different from *Tingia unita* in only possessing scalariform bordered pits on the radial walls of wider tracheid, and 1 to 3 (typically 2 or 3) half bordered or obscurely bordered cross-field pits. Moreover, the multiseriate pits are not observed on the radial tracheid walls of *T. unita*.

*Yangquanoxylon miscellum* is different from *Tiniga unita* in that its scalariform bordered pits are only occasionally present, and its cross-field possesses 1–6 circular bordered pits with oblique elliptical pores, or simple, oval to oblate pits.

In Mesozoic, the scalariform pitting is absent from conifers and Ginkgoales, while it is commonly reported in cycads and Bennettitales including *Scalaroxylon* [1,39], *Chamberlainia* [40], and *Cycadeoidea* [21,41]. However, *Scalaroxylon*, *Chamberlainia,* and *Cycadeoidea* are different from *Tingia unita* due to their manoxylic wood. In addition, there are also some other fossil woods with scalariform pits on radial tracheid walls of secondary xylem, namely *Ecpagloxylon* [42], *Phoroxylon* [43], *Sahnioxylon* [44,45], and *Lhassoxylon* [46]. However, *Ecpagloxylon* is different from *T. unita* in possessing araucarian and mixed type radial tracheid wall pitting, araucarioid cross-field pits, and axial parenchyma cells. *Phoroxylon* is different from *T. unita* in its cross-field with 2 to 6 small, oval, or circular, simple or half-bordered pits. Differing from *T. unita*, *Phoroxylon* has tangential tracheid wall pits. *Sahnioxylon* has uniseriate to multiseriate araucarioid radial tracheid wall pits, uniseriate to quadriseriate rays and araucarioid cross-field pits, so as to be different from *T. unita*. *Lhassoxylon* has axial parenchyma which are absent in *T. unita*.

The morphology and anatomical structure of pith are regarded as the critical diagnostic criteria for the classification and identification of fossil gymnospermous stems, especially to taxonomic identification at generic level. Some trunks with structurally similar secondary xylems are mostly distinguishable based on pith due to its high sensitivity in plant evolution [4,45,46]. Among the above mentioned genera, only half of them has pith reported. *Protopitys buchiana*, *Protopitys scotica,* and *Johnhallia lacunose* are different from *Tingia unita* as their pith only consists of parenchyma cells. *Guizhouoxylon dahebianense* is different from *T. unita* in possessing sclerotic cells, secretory cells, and commissural cells. *Chamberlainia* is different from *T. unita* in having medullary vascular bundles. *Cycadeoidea* is different from *T. unita* in having secretory cells. *Sahnioxylon* is different from *T. unita* in possessing sclerotic nests. It should be noted that *Xuanweioxylon* is very similar to *T. unita* in anatomy. Both of them have secretory canals and residual cell wall bands in pith, endarch primary xylem, low rays, and uniseriate scalariform bordered pits on radial tracheid walls. However, in the pith of *Xuanweioxylon* there are both thin-walled and thick-walled parenchyma cells, while *T. unita* only has thin-walled parenchyma cells. Moreover, the pith septa and scleroid cells of *X. scalariforme* are absent in *T. unita*.

### 4.4. Comparison and Discussion with Other Progymnosperms

Up to now, based on the description and discussion mentioned above, *Tingia* is certainly a progymnosperm due to its gymnosperm-type wood and spore-bearing nature. Here, four orders are recognized as progymnosperms, namely aneurophytales, archaeopteridales, protopityales, and noeggerathians [7,21]. Among them, Aneurophytales is different from *Tingia unita* in which vascular system consists of a lobed or ribbed protostele [21,47,48,49,50]. Archaeopteridales were thought to be closely related to the Noeggerathians based on the plagiotropic anisophyllous pinnules, adaxial attachment of sporangia and heterospory [9,51,52,53]. However, the anatomical information of its stem is quite different from *T. unita*. For example, *Callixylon* as the fossil stem of *Archaeopteris* has eustelic vascular system, one critical feature of *Callixylon* is that the circular bordered pits are grouped together in radially aligned rows and are multiseriate on the radial tracheid walls of secondary xylem, and the adjacent groups of pits are separated by unpitted areas [21,54,55,56]. Whereas, the pitting of *T. unita* is mainly scalariform, uniseriate, and continuous, distributing on the radial tracheid walls of secondary xylem. In addition to *Callixylon*, there are several genera to be assigned to Archaeopteridales, such as *Actinoxylon* [57], *Actinopodium* [58], and *Siderella* [59]. However, they are different from *T. unita* in which vascular strand consists of an actinostele. Protopityales is represented by a single genus and two species, namely *Protopitys buchiana* and *Protopitys scotica* [21], which are also different from *T. unita* based on the above comparison in Section 4.3.

It is widely agreed that the stelar theory has been useful in comparative and phylogenetic studies of fossil vascular plants, especially certain plant groups are characterized by particular stele types [21]. Among the stele types, protostele has been considered as the simplest and oldest stelar type, while the eustele evolved from the continued longitudinal dissection of a protostele [21,60]. As Namboodiri and Beck [60] summarized, the eustele in the gymnosperms has evolved directly from the protostele by gradual medullation and concurrent separation of the peripheral conducting tissue into longitudinal sympodial bundles from which traces diverge radially. So, noeggerathians (at least *Tingia* and *Paratingia*) and protopityales are more evolved than aneurophyton and archaeopteridales in stelar structure. In addition, the primary vascular system has been widely used for systematic purposes due to its relation to leaf trace divergence and anatomy, and the primary xylem is always mesarch in the lobed protostele or actinostele [18]. According to the stellar theory mentioned above, *T. unita* and protopityales are more evolved than aneurophyton and archaeopteridales, so, the endarch primary xylem could be a more evolved feature. However, the primary xylem of protopityales also could be mesarch, so the difference in primary xylem is not a critical diagnostic feature to identify which plant taxa belong to progymnosperms.

During the evolution of wood, the number of pits and the longitudinal columns of pits on the tracheid wall decreased gradually [61]. This evolution trend is also supported by research from other authors [18,22], namely, the pitting of the primitive plant taxa (e.g., aneurophytes) occurs on both radial and tangential tracheid walls, while it is generally restricted on the radial tracheid walls of the derived plant taxa (e.g., seed plants). Considering that the pitting occurred on both radial and tangential tracheid walls of aneurophyton and archaeopteridales, while the pits only occurred on the radial tracheid walls of noeggerathians and protopityales, so noeggerathians and protopityales are relatively evolved.

Both *Tingia* and *Paratingia* mainly possess uniseriate scalariform bordered pits on the radial tracheid walls of secondary xylem. However, the scalariform pitting has been considered as the primitive condition within seed plants, while the circular bordered pits have been considered as evolving from scalariform pits [30,62,63]. In contrast, *Protopitys scotica* possesses multiseriate circular bordered pits on the radial tracheid walls of secondary xylem. At first glance, the stelar structure of protopityales appears to be more evolved than noeggerathians by this theory. But the pitting is mainly uniseriate on the tracheid walls of noeggerathians, while the pitting of protopityales could be multiseriate, which may indicate the noeggerathians is more evolved than protopityales by the theory of the evolution of wood mentioned above. Now we will find there is a contradiction between the two results. Therefore, all of these hypotheses need more evidence to judge.

In general, the anatomical features of *Tingia unita* are unique among the progymnosperms and other genera or species with gymnosperm-typed wood, and the stellar structure of noeggerathiales could be more evolved. Moreover, a further study is necessary for fully understand the relationship between other noeggerathiales and progymnosperms, even with all of other plant taxa.

## 5. Concluding Remarks

To conclude, the stem of *Tingia unita* is eustelar, consisting of a heterocellular pith with parenchyma cells and secretory canals, an endarch primary xylem, pycnoxylic secondary xylem with uniseriate scalariform bordered pits, uniseriate and low rays, parenchymal cortex and fan-shaped or band-like leaf traces. As such, *T. unita* has a combination of gymnospermous wood and heterosporous reproduction [10,11], in accordance with the critical features of progymnosperms [21]. This finding further confirms that Noeggerathiales are progymnosperms, in addition to the evidence in *Paratingia wuhaia* [7]. Furthermore, the stem anatomy characteristics of *T. unita* are different from all of the other reported wood fossils, which means the stem produced by *T. unita* have up to now not been recognized. Finally, compared with a large number of reports on leaf fossils of *Tingia*, the anatomical information of stem of *Tingia* was first reported here.

## Figures and Tables

**Figure 2 biology-12-00494-f002:**
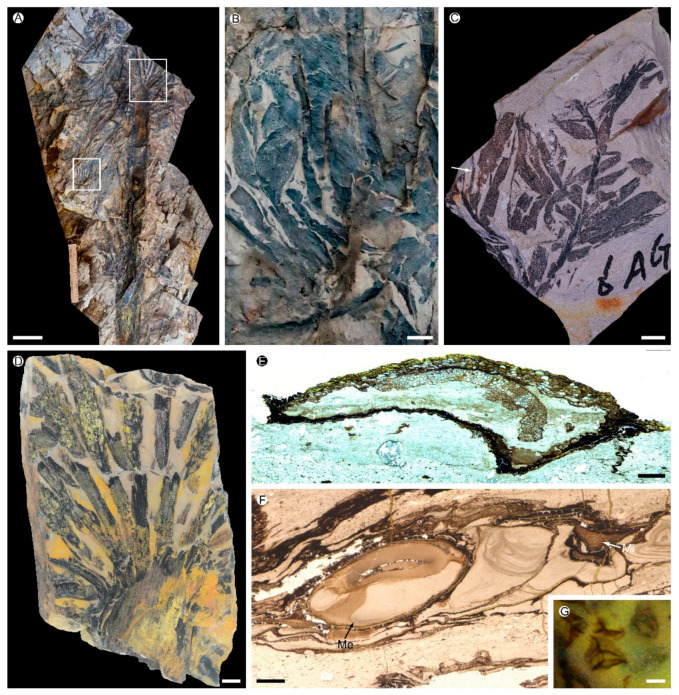
*Tingia unita* from Wuda Coalfield, Inner Mongolia. (**A**) Overall morphology of *T. unita*, both compound leaves and strobili are directly attached on the same stem, scale bar = 10 cm. (**B**) Enlargement of the left white box in (**A**), showing the large pinnules, scale bar = 1 cm. (**C**) Large pinnules and small pinnules (arrow) of *T. uniata*, scale bar = 1 cm. Collection number: PB201694. (**D**) Enlargement of the area within the right white rectangular frame in (**A**), showing the stem part with anatomy examined, scale bar = 1 cm. Collection number: PB201150. (**E**) Showing the C-shaped vascular bundle in rachis of compound leaf, scale bar = 500 μm. Collection number: PB201150; slide number: PB201150-1. (**F**) Showing the megasporangia (Me) and microsporangia (Mi), scale bar = 500 μm. Collection number: PB201150; slide number: PB201150-1. (**G**) Showing the microspores with trilete mark from (**G**) (indicated by the white arrow), scale bar = 10 μm. Collection number: PB201150; slide number: PB201150-1.

**Figure 3 biology-12-00494-f003:**
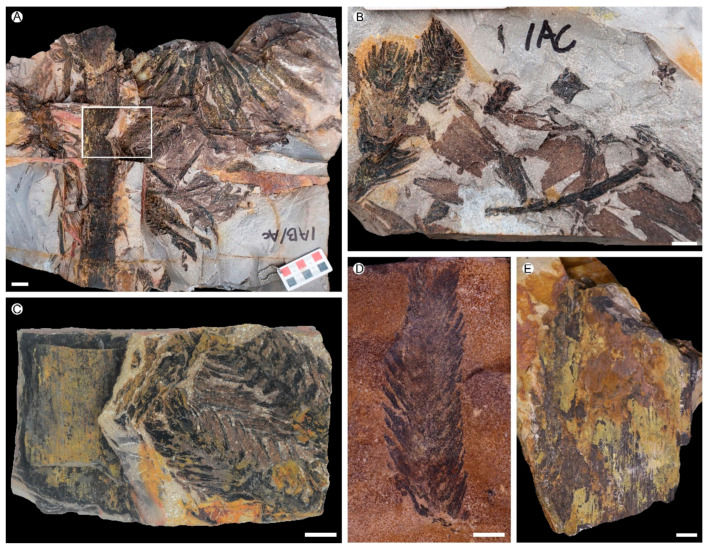
*Tingia unita* from Wuda Coalfield, Inner Mongolia. (**A**) Overall morphology of *T. unita*, both compound leaves and strobili are directly attached on the same stem, scale bar = 2 cm. (**B**) showing the large pinnules and stribilus, scale bar = 1 cm. (**C**) Enlargement of the area within white rectangular frame in (**A**), showing the stem part with anatomy examined, scale bar = 1 cm. Collection number: PB201151. (**D**) Showing the strobili, scale bar = 1 cm. Collection number: PB201695. (**E**) Another part of the stem of *T. unita* was anatomy examined, scale bar = 1 cm. Collection number: PB201152.

**Figure 4 biology-12-00494-f004:**
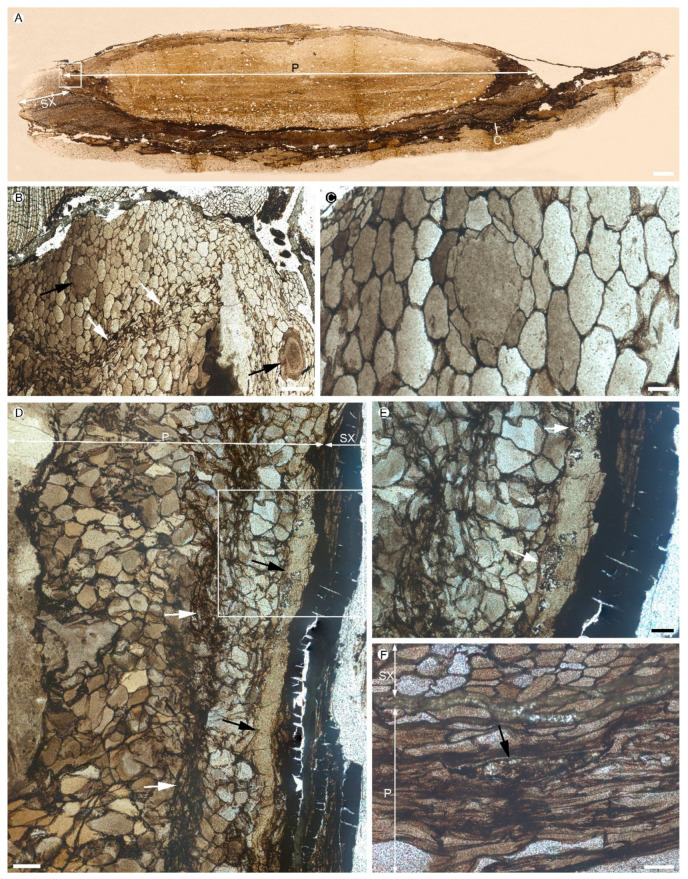
Gross morphology and the pith structures of the stem of *Tingia unita* from the Taiyuan Formation of Wuda Coalfield, Inner Mongolia. (**A**–**C**,**F**) Cross section of the stem. (**D**,**E**) Radial section of the stem. (**A**) Overview of the stem, showing the pith (P), secondary xylem (SX), cortex (C), scale bar = 2 mm. Collection number: PB201152; slide number: PB201152-4. (**B**,**D**) Showing the secretory canals (black arrows) and a narrow band consists of irregularly-shaped parenchymal cells (white arrows). Scale bars: (**B**) = 200 μm; (**D**) = 200 μm. Collection number: PB201152. Slide number: (**B**) PB201152-4; (**D**) PB201152-9. (**C**) Enlargement of the left black arrow in (**B**), showing the secretory canal surrounding by a ring of small cells, scale bar = 50 μm. Collection number: PB201152; slide number: PB201152-4. (**E**) Enlargement of the area within white rectangular frame in (**D**), showing the secretory substance (white arrows), scale bar = 100 μm. Collection number: PB201152; slide number: PB201152-9. (**F**) Showing the secretory substance (black arrow), scale bar = 50 μm. Collection number: PB201150; slide number: PB201150-17.

**Figure 5 biology-12-00494-f005:**
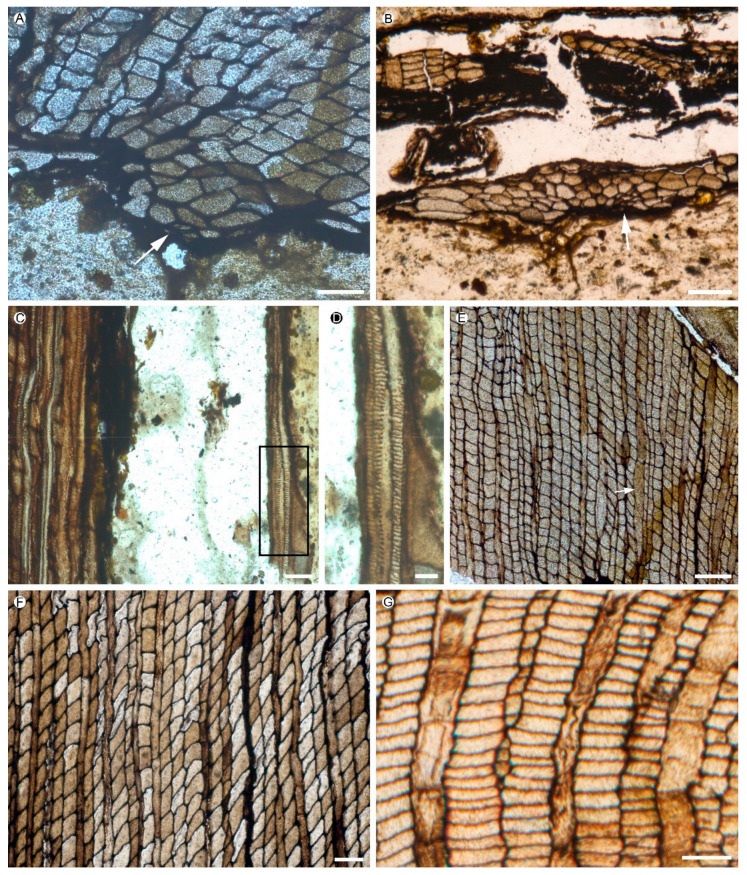
The xylem structure of *Tingia unita* from the Taiyuan Formation of Wuda Coalfield, Inner Mongolia. (**A**,**B**) Cross section of the stem, showing the primary xylem (white arrows). Scale bars: (**A**) = 50 μm; (**B**) = 100 μm. Collection number: PB201150. Slide numbers: (**A**) PB201150-21; (**B**) PB201150-6. (**C**,**D**) Radial section of the stem, showing the helical thickening on the tracheid walls of primary xylem. Scale bars: (**C**) = 50 μm; (**D**) = 20 μm. Collection number: PB201150; slide number: PB201150-11. (**E**–**G**) Cross section of the stems, showing the pycnoxylic secondary xylem with tracheids and rays. Scale bars: (**E**) = 100 μm, (**F**) = 50 μm, (**G**) = 50 μm. Collection numbers: (**E**,**G**) PB201150; (**F**) PB201152. Slide numbers: (**E**) PB201150-4; (**F**) PB201152-3; (**G**) PB201150-5.

**Figure 6 biology-12-00494-f006:**
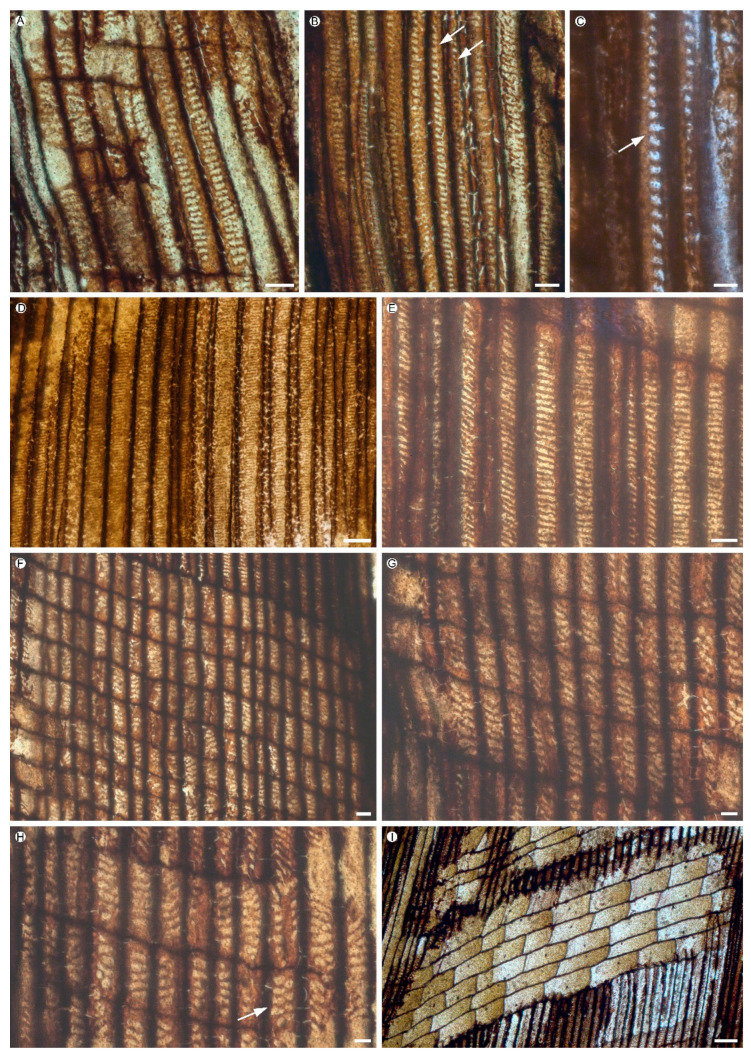
The xylem structure of *Tingia unita* from the Taiyuan Formation of Wuda Coalfield, Inner Mongolia. (**A**–**I**) Radial section of stem. (**A**–**E**) Showing the uniseriate oval (white arrows) and uniseriate scalariform bordered pits on the tracheid walls of secondary xylem. Scale bars: (**A**) = 20 μm; (**B**) = 20 μm; (**C**) = 10 μm; (**D**) = 50 μm; (**E**) = 20 μm. Collection numbers: (**A**–**C**) PB201150; (**D**,**E**) PB201152. Slide numbers: (**A**) PB201150-8; (**B**,**C**) PB201150-11; (**D**) PB201152-9; (**E**) PB201152-12. (**F**–**H**) Showing the uniseriate scalariform to biseriate alternately elliptical (arrow) cross-field pits. Scale bars: (**F**) = 20 μm; (**G**) = 10 μm; (**H**) = 10 μm. Collection number: PB201152, slide number: PB201152-12. (**I**). Showing the ray cells, scale bar = 50 μm. Collection number: PB201152; slide number: PB201152-19.

**Figure 7 biology-12-00494-f007:**
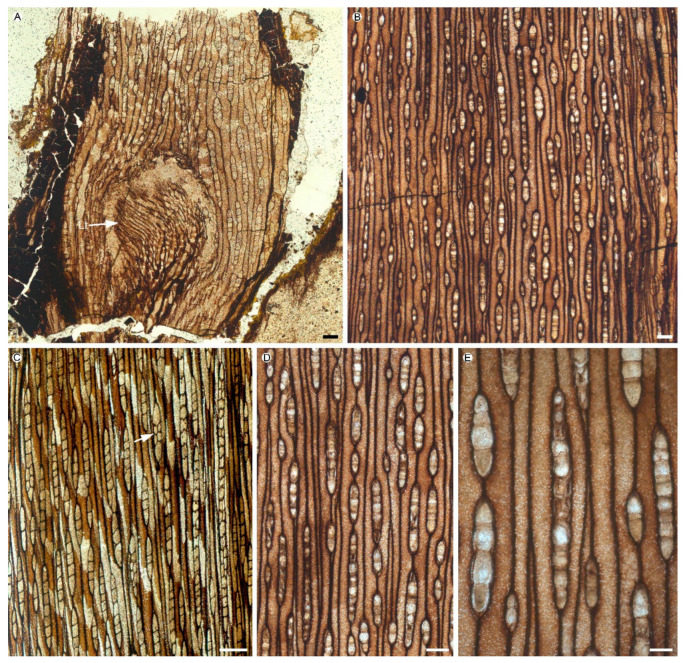
The xylem structure of *Tingia unita* from the Taiyuan Formation of Wuda Coalfield, Inner Mongolia. (**A**–**E**) Tangential section of the stem. Showing the uniseriate to partly biseriate rays. Few ray cells are relatively small than others in the same ray (arrow), and a putative leaf trace (Lt). Scale bars: (**A**) = 100 μm; (**B**) = 100 μm; (**C**) = 100 μm; (**D**) = 100 μm; (**E**) = 50 μm. Collection numbers: (**A**,**B**,**D**,**E**) PB201150; (**C**), PB201152. Slide numbers: (**A**) PB201150-14; (**B**,**D**,**E**) PB201150-16; (**C**) PB201152-22.

**Figure 8 biology-12-00494-f008:**
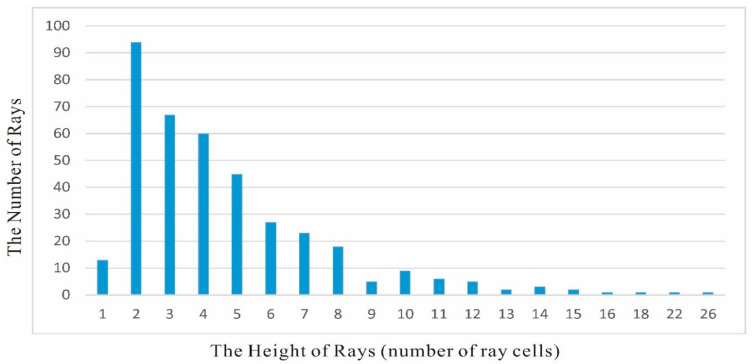
Histogram showing the height-frequency distribution of 790 rays in tangentially longitudinal section of stem.

**Figure 9 biology-12-00494-f009:**
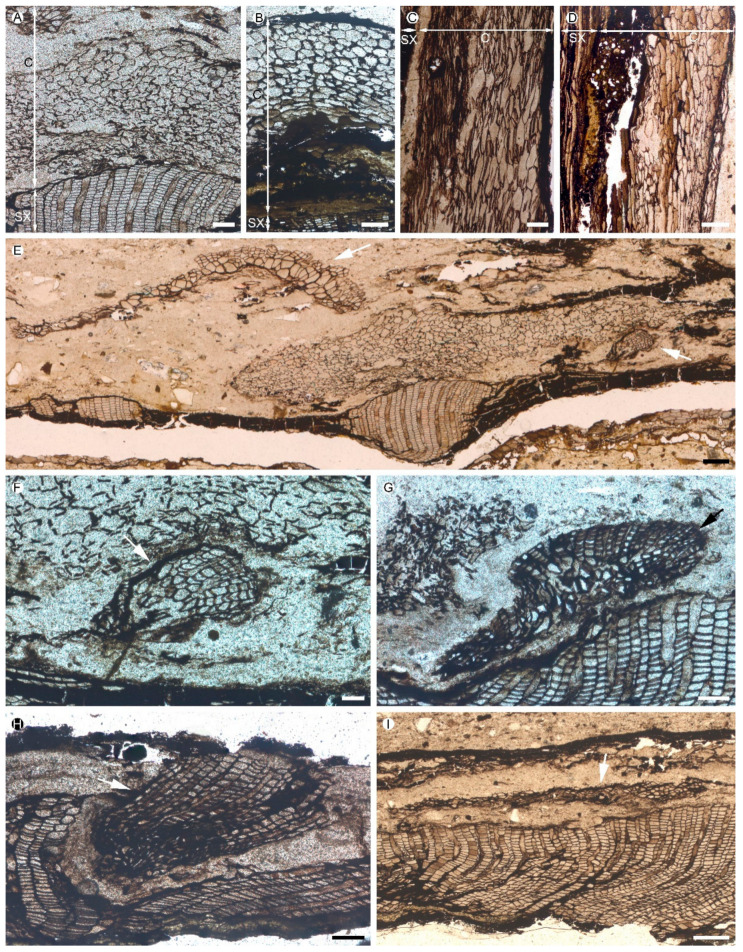
The cortex and leaf traces structures of *Tingia unita* from the Taiyuan Formation of Wuda Coalfield, Inner Mongolia. (**A**,**B**) Cross section of the stem, showing the secondary xylem (SX) and cortex (C). Scale bars: (**A**) = 100 μm; (**B**) = 200 μm. Collection numbers: (**A**) PB201150; (**B**) PB201152. Slide numbers: (**A**) PB201150-5; (**B**) PB201152-1. (**C**,**D**) Radial section of stem, showing the longitudinally elongated parenchyma cells in the cortex. Scale bars: (**C**) = 200 μm; (**D**) = 200 μm. Collection numbers: (**C**) PB201150; (**D**) PB201152. Slide numbers: (**C**) PB201150-9; (**D**) PB201152-20. (**E**–**I**) Cross section of the stem, showing the leaf traces (arrows) in different shape. Scale bars: (**E**) = 200 μm; (**F**) = 50 μm; (**G**) = 100 μm; (**H**) = 100 μm; (**I**) = 200 μm. Collection number: PB201150. Slide numbers: (**E**,**F**,**I**) PB201150-5; (**G**) PB201150-6; (**H**) PB201150-25.

## Data Availability

All data generated by this study are available in this manuscript.

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
