# Peer review of "Stem Anatomy Confirms Tingia unita Is a Progymnosperm"

_biology, 2023, doi:10.3390/biology12040494_

Round 1

Reviewer 1 Report

Based on stem anatomy, this paper confirms that Tingia unita is a progymnosperm and the Noeggerathiales is included in the progymnosperms. The anatomy provided is in detail, the conclusion is acceptable and the work is very important. I recommend publication of the paper after addressing the following comments.

The authors need to explain what are pycnoxylic and manoxylic secondary xylems (woods) and what groups of plants possess such xylems; and what is cross-field. They should also provide or point to figures to make sure that the pits on tangential tracheid walls are absent.

In progymnosperms, e.g., Aneurophytales, the bordered pits occur on both radial and tangential walls of the secondary xylem tracheids. How about Archaeopteridales? So, why Tingia unita as a progymnosperm has pits only on radial walls of the secondary xylem tracheids? As a derived character, such occurrence of pits typifies seed plants (see Wang and Liu, 2015. A new Late Devonian genus with seed plant Affinities. BMC Evolutionary Biology (2015) 15:28; Galtier J., 1988. Morphology and phylogenetic relationships of early pteridosperms. In: Beck CB, editor. Origin and evolution of gymnosperms. New York: Columbia University Press; p. 135–76).

Progymnosperms (Aneurophytales and Archaeopteridales) are characterized by mesarch primary xylem. However, Tingia unita has endarch primary xylem. This must be discussed in the paper.

In the abstract, the authors say “The stem of Tingia unita consists of parenchymatous pith, endarch primary xylem, pycnoxylic secondary xylem and cortex, well accordant with a typical stem of a gymnosperms. As a result, Tingia unita is certainly a progymnosperms.”. Why the authors transfer a gymnosperm to a progymnosperm, based on the same anatomy and expression?

In the title, abstract and throughout the manuscript, a progymnosperms should be replaced by a progymnosperm. Other problems in expressions and grammar should be resolved (I just give some examples, see below).

Line 20, evidences is right?

Lines 50, 60, why Figure 2 rather than Figure 1 first appears in the paper?

Line 54, strengthen or strengthens?

Line 77, in Figure 1A, is a scale necessary?

Line 109, the enlarged area in B does not appear exactly matching the area in white box of A.

Lines 155-156, scalariform bordered pits are not very clear on the tracheid walls of secondary xylem. Some areas of the figure need enlargement.

Lines 179-180, is it true that the middle part of rays is narrower than both ends? the width of rays are gradually increase…should be corrected.

Line 184, where is the evidence that pits on tangential tracheid walls are absent?

Line 202, is it right to say It is worthy of mention?

Lines 214-215, is it right to say Sphenophyllum has protostele vascular cylinder with the primary xylem is…?

Lines 236-237, is it right to say there are some other species also possesses?

Line 298-299, why parenchyma cells differentiate into thin-walled and thick-walled cells?

Line 309, the current stem represents a new type of fossil stem. What is the exactly new type?

Line 327, at this reviewing stage, why the authors know there are four reviewers?

Author Response

Thanks for your comment and suggestion, they are very important to our manuscript. We have revised the manuscript. Please see the attachment to get the reply .

Thanks you again.

Reviewer 2 Report

Tingia, as one of the index plants in the Order Noeggerathiales, is very important in denoting its affinite among the other Paleozoic extinct taxa, although the anatomy of Paratingia was revealed to be gymosperm wood with heterosporous repruductive organs. The present materials of the stem anatomy of Tingia would be interesting for both botanists and paleobotanists and the general readers study the land plants evolution. Therefore, I suggest to publish this work on the  journal Biology soon. Howerver, there are some points still need to be improved as the attached review file.

First of all, we need to prove what we discussed herein this manuscript belongs to the species T. unita. More words might be needed in the description and systematic comparision section.

Second, more detailed pictures are needed to the key characteristics of gymnosperm wood, such as the bordered pits on the tracheids, eustele, endarch xylem.

Third, a small pragraph might be needed to discuss the botanical affinity with other known progymnospers, such as Archaeopteridales, Protopityales, etc.

Fourth, here we follow the whole plants and named the stem after  T. unita, However, there might be dispersed woods found sooner or later, it is possible to give an indepandant genus and species names, and provide the diagnostics on genus and species level respectivrly.

Fifth, the canal is not so important to this article, the caption of Figure 3 is very confused. Fig 3C does not look like a canal, could you please provide more pictures of canals. For the so-called canal in Fig 5F is also suspicious. For Fig 5E, could you please provide some close-ups of the canals. The so-called irregularly-shaped parenchymal cells might be resulted by deformation during preservation?

Finally, the description should be improved, more photos should be provided. such as: 

1.      Line 151, “Rays are parenchymatous, and the tangential width of ray cells is usually narrower than that of tracheid in cross section.” What is the meaning of “that”, the tangential width of tracheid??

2.      Line 152, “Occasionally, some ray cells are wider than the tracheids in tangential direction.” I can not understand, please indicate the photos showing this character.       In radial section, It seems that the pits and reticulate thickenings are present at the same time. The cross-field pits seem present in some units of Fig 4A. The authors should provide more photos.

Author Response

Thanks for your comment and suggestion, they are very important to our manuscript. We have revised the manuscript. Please see the attachment to get the reply .

Thanks again.

Reviewer 3 Report

This is a good and convincing description of the woody anatomy of Tingia and should be published with minor revision

My main misgiving is that it is never explained why the wood belong to Tingia: a genus based on leaves and cones. I gather that the reason is attachment, but the attachment of the cones to the large trunks is by no means clear. It there a thin section of the cone peduncle? Nor is there any indication of leaves. I think a paragraph detailing the basis of this whole plant reconstruction is needed, including explanation why it could not be wood of Alethopteris, Taeniopteris or Pterophyllum in the same deposit. I do agree that there is the appearance of whole plants overwhelmed by ash, and this should be a component of the argument.

The title (also l.30) needs correction: should be a progymnosperm (singular)

l.38 rather than “apart from”, would not in “addition to” be better

l.45 rather than “has got” use is “known for”

l.49 not “petrified”, ie turned into stone or agatized but “permineralized” and retaining cell structure

Subheadings 3.2.1 and other should be capitalized “Leaf” not “leaf”

Author Response

(The authors gave the same response as above.)

Reviewer 4 Report

Please find the comments in the attachment.

Author Response

(The authors gave the same response as above.)

Round 2

Reviewer 1 Report

I think the paper has been greatly improved. The authors should still pay attention to spelling and grammar somewhere.

Reviewer 2 Report

Although the geological age of the horizon where the fossil woods in situ preserved assigned to the earliest Permian based on the U-Pb isotopic dating , the biostratigraphic correlation still shows that this horizon might be Artinskian. However, this is not the main topics of the present paper.